# A Correlational Analysis of Phthalate Exposure and Thyroid Hormone Levels in Common Bottlenose Dolphins (*Tursiops truncatus*) from Sarasota Bay, Florida (2010–2019)

**DOI:** 10.3390/ani12070824

**Published:** 2022-03-24

**Authors:** Miranda K. Dziobak, Randall S. Wells, Emily C. Pisarski, Ed F. Wirth, Leslie B. Hart

**Affiliations:** 1Environmental and Sustainability Studies Graduate Program, College of Charleston, Charleston, SC 29424, USA; 2Environmental Health Sciences Graduate Program, University of South Carolina, Columbia, SC 29208, USA; 3Chicago Zoological Society’s Sarasota Dolphin Research Program, c/o Mote Marine Laboratory, Sarasota, FL 34236, USA; rwells@mote.org; 4National Oceanic and Atmospheric Administration, National Ocean Service, National Centers for Coastal Ocean Science, Charleston, SC 29412, USA; emily.pisarski@noaa.gov (E.C.P.); ed.wirth@noaa.gov (E.F.W.); 5Department of Health and Human Performance, College of Charleston, Charleston, SC 29424, USA

**Keywords:** endocrine disruption, thyroid hormone, cetacean, phthalates

## Abstract

**Simple Summary:**

Phthalate exposure is prevalent in common bottlenose dolphins sampled from Sarasota Bay, Florida. With evidence of potential adverse effects as identified in human and laboratory studies, there is a concern for bottlenose dolphin health. This study investigated potential correlations between serum hormone levels and urinary phthalate metabolite concentrations to begin to understand whether health effects would be expected in dolphins. We observed a positive relationship between free thyroxine and mono(2-ethylhexyl) phthalate (MEHP) for both adult female and male dolphins, suggesting potential associations with normal thyroid production.

**Abstract:**

Phthalates are chemical esters used to enhance desirable properties of plastics, personal care, and cleaning products. Phthalates have shown ubiquitous environmental contamination due to their abundant use and propensity to leach from products to which they are added. Following exposure, phthalates are rapidly metabolized and excreted through urine. Common bottlenose dolphins (*Tursiops truncatus*) sampled from Sarasota Bay, Florida, have demonstrated prevalent di(2-ethylhexyl) phthalate (DEHP) exposure indicated by detectable urinary mono(2-ethylhexyl) phthalate (MEHP) concentrations. Widespread exposure is concerning due to evidence of endocrine disruption from human and laboratory studies. To better understand how phthalate exposure may impact dolphin health, correlations between relevant hormone levels and detectable urinary MEHP concentrations were examined. Hormone concentrations measured via blood serum samples included triiodothyronine (T3), total thyroxine (T4), and free thyroxine (FT4). Urinary MEHP concentrations were detected in 56% of sampled individuals (*n* = 50; mean = 8.13 ng/mL; s.d. = 15.99 ng/mL). Adult female and male FT4 was significantly correlated with urinary MEHP concentrations (adult female Kendall’s tau = 0.36, *p* = 0.04; adult male Kendall’s tau = 0.42, *p* = 0.02). Evidence from this study suggests DEHP exposure may be impacting thyroid hormone homeostasis. Cumulative effects of other stressors and resultant endocrine impacts are unknown. Further research is warranted to understand potential health implications associated with this relationship.

## 1. Introduction

Phthalates are a class of man-made chemical esters commonly used as plasticizers to increase the flexibility and durability of polyvinyl chloride (PVC) [1]. As an extremely versatile additive, phthalate use is not limited to PVC but includes a myriad of consumer goods such as personal care products, packaging, agricultural insecticides and pesticides, and medical devices [1,2,3]. Phthalates can easily leach from the products to which they are added [4,5], providing abundant exposure opportunities for humans and wildlife. Human exposure can occur via dermal absorption [6], ingestion [7], inhalation [8], or through intravenous medical device usage [9]. Following exposure, phthalate parent compounds are rapidly metabolized and excreted through urine and feces as metabolites [10,11]. As a result, phthalate exposure is signaled by detectable phthalate metabolite concentrations.

Phthalates are considered endocrine-disrupting chemicals (EDCs) and have been linked with altered hormone concentrations, including reproductive (e.g., reduced testosterone and elevated progesterone [12,13,14]), adrenal (e.g., reduced cortisol and aldosterone [15,16]), and thyroid hormones (e.g., reduced triiodothyronine and free thyroxine [16]). Laboratory rodent studies and human observational studies have demonstrated that endocrine disruption can vary and is likely dependent on a number of factors, including phthalate ester type [17], dose [18], exposure window [19], species [20], sex [21], age [22], and pregnancy status [23]. Toxin metabolism generally occurs in at least two steps: phase I hydrolysis followed by phase II conjugation [24]. Long-branched phthalates, such as di-(2-ethylhexyl) phthalate (DEHP), may undergo further hydroxylation and oxidation in order to be excreted [24]. The metabolic process involves the peroxisome proliferator-activated receptor alpha (PPAR-α), which can be activated by mono(2-ethylhexyl) phthalate (MEHP) [25]. PPARs are nuclear receptors responsible for cholesterol uptake and transport; so, proliferation may play a role in downregulated steroidogenesis [26,27,28].

Evidenced in epidemiological and laboratory studies, this endocrine disruption may result in male reproductive abnormalities (e.g., malformations of external genitalia [29,30] and altered sperm function [31,32]), early pregnancy loss [33], disrupted male sexual differentiation [29], increased breast cancer risk [34], increased oxidative stress [35], and adverse effects on growth and development [36]. The timing of exposure (e.g., pre/postnatal) seems to be an important determinant in the severity of some phthalate-mediated health impacts; previous studies have linked phthalate exposure with severe fetal neurodevelopmental impacts [22,37]. Similarly, gestational, and perinatal phthalate exposure may have long-term health implications, even after exposure has ceased [38,39]. Previous phthalate studies have focused on humans and rodents; so, health effects detected in other higher-order mammalian species are not well studied.

Recently, prevalent phthalate exposure has been detected in free-ranging common bottlenose dolphins (*Tursiops truncatus*) that are long-term residents of Sarasota Bay, Florida (i.e., ~75% of sampled individuals had detectable urinary concentrations of at least one phthalate metabolite [40,41,42]). Dziobak et al. (2021) did not find demographic-dependent susceptibility to exposure, and some metabolite concentrations exceeded levels reported for human reference populations [42]. Higher dolphin MEHP concentrations were notable considering humans are exposed to phthalates through the active use of phthalate-containing products, while dolphins are exposed due to environmental contamination. Specific sources and persistence of phthalate exposure are still unknown; but, given that DEHP exhibits low water solubility and preferentially partitions to suspended particles and sediments in water [1,43], dolphin exposure could rely on food-based sources (e.g., consuming microplastic-contaminated prey that may release phthalates [44]). The health impacts from this heightened exposure in dolphins are currently unknown. Epidemiological studies have been conducted to understand relationships between marine mammal health impacts and exposure to polychlorinated biphenyls (PCBs) [45], organochlorine pesticides (OCPs) [45], and polybrominated diphenyl ethers (PBDEs) [45], thus providing a framework for the exploration of hormonal correlates with phthalate exposure. Unlike these environmentally persistent chemicals that have been observed to bioaccumulate in dolphin tissue, phthalates’ metabolism is expected to occur rapidly [11]; however, the exact mechanism in dolphins is unknown.

The ongoing release of phthalates into the environment [46] presents a chronic exposure risk for this community. Given the high prevalence of MEHP exposure in dolphins, MEHP concentrations that exceed human reference populations, and the potential for phthalate-induced endocrine disruption, the objective of this study was to examine the relationship between urinary MEHP concentrations and a suite of hormones (adrenal, reproductive, and thyroid) in free-ranging bottlenose dolphins. Chronic phthalate exposure has been observed to negatively impact human health; so, similar health impacts may be expected for Sarasota Bay dolphins as well. Findings from this study can be used to better understand the extent of estuarine phthalate contamination and potential health risks to exposed wildlife.

## 2. Materials and Methods

### 2.1. Dolphin Community and Sample Collection

Dolphins sampled for this study were individuals considered to be members of the year-round, multi-decadal, multi-generational resident community in Sarasota Bay, FL, USA (n~160) [47]. High site fidelity has enabled long-term research efforts regarding the life history, behavior, and health of this population [48,49]. Urine and blood samples for this study (2010–2019) were collected during routine, periodic, catch-and-release health assessments [48,50] conducted under Scientific Research Permits #522-1785, #15543, and #20455 from the National Oceanic and Atmospheric Administration’s (NOAA) National Marine Fisheries Service (NMFS). All catch-and-release and sampling methodologies for the health assessments were reviewed and approved annually by Mote Marine Laboratory’s Institutional Animal Care and Use Committee (IACUC). Blood was drawn prior to other biological sampling, so all individuals sampled for urine had corresponding blood samples. Some dolphins (*n* = 13) were sampled more than once, but the analyses conducted herein relied upon the most recently obtained sample.

Blood and urine collection methods have been previously described [48,51,52]. Briefly, blood was drawn from the ventral fluke via butterfly catheter into serum separator tubes. Serum samples were kept at room temperature for 45 min before being centrifuged and frozen in liquid nitrogen in preparation for overnight shipment. Samples were shipped to Cornell University’s Animal Health Diagnostic Center’s (AHDC) Endocrinology Laboratory (Ithaca, NY, USA) for hormone analyses. Following blood draw, dolphins were brought aboard a specially designed veterinary examination vessel where urine was opportunistically collected. Standardized urine collection methods as previously reported [41,50] involved insertion of a catheter (Kendall Sovereign Feeding Tube and Urethral Catheter 8Fr/Ch × 22 in 27 mm × 56 cm, Covidien, Dublin, Ireland) coated with a sterile surgical lubricant (Surgilube^®^, Stoelting, Wood Dale, IL, USA) into the urethra by a trained veterinarian.

### 2.2. Sample Processing and Analysis

Analysis, quantification, and quality control methods for urinary phthalate metabolite screening have been described [40,41], and were based on protocols established by the Centers for Disease Control and Prevention (CDC). Individual urine samples were analyzed in batches, and quality assurance/quality control (QA/QC) samples (reagent blanks, field blanks, reagent spikes, matrix spikes, and SRM 3672 Organic Contaminants in Smokers’ Urine) were processed concurrently [40,41]. Sample integrations were performed using Analyst software (ver 1.5, SCIEX, Framingham, MA, USA). MEHP, the first metabolite of di(2-ethylhexyl) phthalate (DEHP; ATSDR, 2019), was the most frequently reported metabolite by Dziobak et al. (2021) and used for analyses herein. Serum sent to AHDC was analyzed for triiodothyronine (T3), total thyroxine (T4), and free thyroxine (FT4) concentrations. Hormone analyses using Siemens Immulite Total T3 Chemiluminescent Assay (Gwynedd, United Kingdom; LOD = 19 ng/dL), Siemens Immulite Total T4 Chemiluminescent Assay (Gwynedd, United Kingdom; LOD = 0.30 ug/dL), and Antech Free T4 by Dialysis Radioimmunoassay (Irvine, CA, USA; LOD = 0.15 ng/dL) kits were performed by one laboratory at Cornell University as inter-laboratory variations can be significant [51].

### 2.3. Statistical Methods

Descriptive statistics were used to summarize phthalate metabolite concentrations and measured hormone levels overall and by sex and age class. Dolphins were classified as juveniles or adults based on sexual maturity status as determined by several factors including age, pregnancy diagnosis, calving history, and sex hormone concentrations [41]. Thyroid hormone values lower than the limit of detection (LOD) were set to zero [53], and phthalate metabolite data as reported in Dziobak et al. (2021) were used for descriptive, bivariate, and correlational analyses. Censored analyses were performed for any analyte where at least 20% of the values were below LOD to compare demographically and generate Kendall’s tau (NADA2 R package, R Foundation for Statistical Computing, Vienna, Austria) [54]. A Shapiro–Wilk test was used to evaluate the Gaussian distribution of each analyte. For analytes that were distributed normally, an independent t-test was used to compare concentrations demographically. Censored analytes were compared demographically with a permutation test of differences [55]. Samples from pregnant dolphins were excluded from analysis [56,57] (*n* = 1).

Demographic stratification strategies were determined by generalized linear modeling (GLM), which identified significant relationships with sex, age class, and the interaction of sex and age class. Relationships between MEHP and thyroid hormones were analyzed using the Akritas–Theil–Sen (ATS) line for censored data to compute Kendall’s tau correlation coefficient and *p*-value. Following previously reported values [40,41,42], sample normalization (via creatinine or specific gravity) was not conducted prior to quantification. All statistical analyses were conducted using Statistica (Version 13, Dell Inc., Round Rock, TX, USA) and R (Version 3.2, R Foundation for Statistical Computing, Vienna, Austria) software packages. Statistical significance was evaluated using α = 0.05.

## 3. Results

Excluding repeated sampling events and pregnancies, 50 matched serum and urine samples were collected from unique Sarasota Bay dolphins during 2010–2019 (female *n* = 29; male *n* = 21; adult *n* = 33; juvenile *n* = 17). A total of 56% of individuals demonstrated detectable urinary MEHP concentrations. Adult females (*n* = 29) had the highest MEHP detection frequency (66.67%), and adult males (*n* = 21) had the lowest (43.75%; Appendix A). T4 and FT4 were detected in 100% of individuals (Table 1).

MEHP and thyroid hormone concentrations are summarized by sex and age class in Table 2 and Table 3, respectively. A permutation test of differences found no variation in MEHP concentrations between sexes (*p* = 0.16) or age classes (*p* = 0.25). Results from the GLM identified a significant association between age class and T4 (*p* < 0.0001), as well as a significant association between both sex and age class parameters and FT4 (*p* < 0.05; Table 4). The interaction of sex and age class was not significantly associated with any thyroid hormone.

In adult females and males, FT4 exhibited a significant positive correlation with MEHP (Figure 1 and Figure 2). The remaining hormones, T3 and T4, were not significantly associated with MEHP, regardless of the stratification method (Appendix A).

## 4. Discussion

### 4.1. Overall Findings

To our knowledge, this is the first study investigating phthalate-associated endocrine disruption in free-ranging dolphins. We examined relationships between thyroid hormones and urinary MEHP concentrations in Sarasota Bay dolphins sampled during 2010–2019. FT4 values reported in this study were similar to values previously reported for dolphins sampled from Sarasota Bay [56,58]. FT4 values from this study were also similar to ranges reported for dolphins sampled near Charleston, South Carolina [59]; however, values from different locations may not be comparable. Thyroid hormone concentrations can vary based on the specific conditions present at each location (e.g., water temperature, prey availability, and nutritional quality of prey [58,59]). FT4 was found to be significantly related to MEHP for both adult females and males, suggesting a potential role in thyroid homeostasis. In humans, MEHP is generally inversely associated with FT4 levels [16,23]; however, some studies of female children have found a positive relationship [21,60]. In fact, Weng et al. (2017) reported similar results to this study, where MEHP was positively associated with FT4 in girls, but was not associated with T3 or T4. Disrupted thyroid function appears to be more prevalent in women compared to men [61], potentially related to the sex-specific regulation of thyroid hormones in the brain [62]. Still, significant associations between MEHP and FT4 have been demonstrated in adult male humans [16]. Health effect differences reported in human studies could be due to phthalate metabolism differences. Compared to adults, children have reduced toxin-metabolizing enzymes and diminished renal excretion capacity [63,64], which could affect their ability to metabolize phthalates. In fact, children have shown higher excretion levels of oxidized DEHP metabolites (e.g., mono(2-ethyl-5-oxo-hexyl) phthalate (MEOHP) and mono(2-ethyl-5-hydroxyhexyl) phthalate (MEHHP)) than adults, implying age-related differences in phthalate elimination routes [65,66].

Phthalate metabolism is likely different between humans and dolphins as well; Kluwe (1982) observed variations in DEHP metabolism among terrestrial mammals, and differences in the ability to metabolize other common marine contaminants (e.g., PCBs) have been observed in marine–terrestrial mammal comparative studies [67]. Thus, it is reasonable to expect marine mammal differences in phthalate metabolism. Additionally, measured thyroid weight-to-body weight ratios are reportedly higher in dolphins than terrestrial mammals [68], further suggesting the potential for metabolic differences. It is important to consider metabolic capabilities with regards to expected health outcomes because different chemical forms exhibit different bioactivities. Metabolism is generally known to detoxify and facilitate excretion of xenobiotics; however, DEHP metabolism has been shown to increase toxicity as MEHP is the more bioactive form [69,70]. As a result, it might be expected that differing metabolic capabilities would result in different associated health impacts as well. Further research is needed to understand whether the determined association between MEHP and FT4 is impacting dolphin health.

### 4.2. Mechanisms of Disruption

The mechanisms of DEHP-induced thyroid disruption are not well understood. The thyroid-stimulating hormone (TSH) mediates thyroid hormone release by stimulating the thyroid gland [71]. Previous studies have linked DEHP exposure with thyroid gland hyperactivity in that exposure was associated with increases in T3 and T4 [72] as well as physical changes to the gland consistent with hyperactivity (e.g., shrunken colloid, hypertrophy of the Golgi apparatus, and dilation of the rough endoplasmic reticulum [73]). While not completely understood, there is evidence from zebrafish models suggesting that DEHP may upregulate mRNA expression of the TSH gene, resulting in increased T4 levels [74]. Since TSH can initiate T4 synthesis, perhaps the observed increase was a direct result of abnormal gene expression [74] The relationship we found between MEHP and FT4 in this study may also be due to interferences with transthyretin (TTR). TTR is a transport protein mainly responsible for binding and transporting T4 [75]. DEHP exposure may inhibit TTR internalization and expression [76] or competitively bind to TTR [77,78], resulting in higher levels of FT4. TSH is sensitive to minor changes in thyroid hormones; so, future research should examine relationships between MEHP and TSH.

### 4.3. Implications of Thyroid Dysfunction

Thyroid hormones, through regulation of growth, thermogenesis, and metabolism, affect virtually every organ system [79]. As a result, deviations from normal levels can significantly impact overall health. Hyperthyroidism, characterized by increased thyroid levels, has been associated with dementia [80], all-cause mortality [81], and frailty in adult men (i.e., multiple organ system deterioration that leads to diminished capacity to cope with stressors, and increased risk of death and disability [80]) as well as bone deterioration in postmenopausal women [82]. Non-sex-specific associations with hyperthyroidism include increased low-density lipoprotein cholesterol [83] and thyroid cancer [84,85]. It is not understood how the relationship between adult FT4 and MEHP will affect dolphin health, but outcomes observed in hyperthyroidic humans raise concerns. While adult thyroid disruption can lead to serious consequences, related health impacts may be more severe for developing fetuses. Fetuses do not produce their own thyroxine and, instead, rely on maternal inputs to maintain normal thyroid function [86,87]. In humans, maternal hyperthyroidism can result in reduced fetal growth, low birth weight, and even fetal death [88]. Currently, the broader health impacts of hyperthyroidism are unknown for bottlenose dolphins; but, in harbor seals, hyperthyroidism has resulted in significant impacts to energy metabolism during diving, such as increased post-dive lactate concentrations and decline in heart rate [89]. For bottlenose dolphins, additional research on impacts from thyroid imbalance is warranted.

### 4.4. Triiodothyronine (T3)

We did not find significant correlations between MEHP and T3. Given the relationship with FT4, the lack of correlation between MEHP and T3 was surprising. Previous contaminant studies have demonstrated concurrent associations with T3 and FT4 [16,58]. While the thyroid gland releases some T3, the majority is produced through T4 deiodination [71,90]. As a result, it would be expected that an association between MEHP and FT4 would also be observed with T3; however, this was not the case with dolphins. This could be a result of sample size limitations as not every dolphin had detectable T3 levels. A post hoc power analysis was conducted and found power for the correlations was less than the 80% threshold previously determined [91] (Appendix A). A larger sample size is warranted to establish whether the findings from this study are true. It is also possible that feedback mechanisms regulating hormone production prevented MEHP-related fluctuations in T3. Thyroid hormone signaling is substantially modulated by deiodinases that enzymatically activate or deactivate thyroid hormones [92,93]. In cases of increased T4, deiodinases can inactivate T3 as well as prevent further activation of T4, thus providing some measure of protection against hyperthyroidism [94].

While sample size and biological or physiological differences between bottlenose dolphins and humans may partially explain deviations from laboratory and human epidemiological findings, extrinsic factors, such as exposure to contaminant mixtures, should also be considered. For example, Sarasota Bay dolphins are exposed to a mixture of toxins and toxicants related to thyroid function (PCB mixtures [45,49]); but, we are uncertain how endocrine homeostasis is impacted by chemical mixtures. Further study is warranted to investigate synergistic or antagonistic impacts of DEHP exposure and concurrent exposure to other environmental contaminants. 

### 4.5. Strengths and Limitations

This was the first investigation to correlate urinary phthalate metabolite concentrations with circulating hormone levels in dolphins, providing insight into potential health effects posed by phthalate exposure. Long-term monitoring programs and periodic health assessments conducted in Sarasota Bay facilitated biological sample collection over a 10-year sampling period, which helped to maximize sample sizes for correlation assessments by key demographic factors. This stratification is necessary for analyses of sex- and age-dependent hormones. We used matched samples so that urinary phthalate metabolite and hormone concentrations were obtained from the same dolphin. Additionally, we employed statistical methods that included values below LOD to examine hormonal relationships with dolphins that had little to no indication of phthalate exposure. Seasonality can significantly alter circulating thyroid hormone concentrations, where decreased concentrations of thyroid hormones have been linked with increased water temperatures [59]. The majority of samples from this study were collected in May, with some sampling occurring in June and July. Water temperature from May to July typically ranges from 79° F to 86° F [95]; so, significant, seasonal fluctuations in thyroid hormones would not be expected.

This study relied on opportunistic sampling of wild dolphin urine and blood, limiting our ability to select individuals based on desired demographics. As a result, we had an uneven distribution of individuals between age classes, which may have limited our ability to detect correlations when age class was used as a stratification method, particularly for juvenile dolphins. While phthalates are not expected to be stored in tissue [11], recent studies have provided some evidence for phthalate metabolite detection in marine mammal blubber (e.g., MEHP detected in harbor porpoises, *Phocoena phocoena* [96], and fin whales, *Balaenoptera physalus* [97]). As such, dolphin body condition may play a role in phthalate bioavailability for metabolism; however, further research is needed to understand this potential relationship. Additionally, our analyses only focused on linkages between hormones and phthalate exposure; however, Sarasota Bay dolphins are also exposed to other toxins and toxicants with endocrine-disrupting effects (e.g., PCBs [45,49,98], hexachlorobenzene (HCB) [98], mercury [99], OCPs [45], and PBDEs [45]). Potential cumulative and interactive effects between phthalates and other contaminants are not yet understood for bottlenose dolphins; however, multiple stressors can interact additively, synergistically, or antagonistically, and potentially result in significantly different effects [100]. Relationships between circulating hormone levels and chemical contaminant detection in dolphins have been reported for PCBs [53,58], DDTs [53], chlordane [53], mirex [53], dieldrin [53], HCB [53], and brominated diphenyl ethers (BDEs) [53]. As many of these chemicals have also been detected in Sarasota Bay dolphins, phthalate exposure may be one of many influences on endocrine function. For example, significant interactive effects have been reported between PCBs and DEHP, resulting in reproductive impairment in humans, dogs, and rats [101,102]. As such, interactive effects between PCBs and DEHP may be observed in dolphins as well. Future studies should seek to identify the pollutant mixtures in Sarasota Bay and examine relationships between determined contaminants and circulating hormone levels to further elucidate potential mechanisms of hormone impairment. The relationship between FT4 and MEHP reported in this study may be true; however, other factors, such as diet, have also been shown to alter FT4 concentrations. Although not statistically significant, a decrease in bottlenose dolphin FT4 has been observed during fasting, followed by an increase in FT4 up to 11 h after re-feeding as compared to baseline levels [103]. As a result, there is the potential for the observed relationship to be associated with factors other than phthalate exposure.

## 5. Conclusions

Potential health risks posed by phthalate exposure are complex and rely on numerous factors to determine specific endpoints. Exposure to a chemical can influence disease processes through multiple mechanisms of action; so, it is important to investigate multiple avenues of potential health effects. Bottlenose dolphins from Sarasota Bay, Florida, were evaluated for relationships between detectable phthalate metabolite concentrations and circulating hormone levels. In adult female dolphins, FT4 was found to have a significant relationship with MEHP. Health effects resulting from this association are currently unknown; so, further monitoring of thyroid hormone levels is warranted. Hyperthyroidism indicated by elevated FT4 levels has been associated with weight loss in humans and dogs [104,105]; so, monitoring dolphin body condition may provide preliminary insight into potential thyroid disruption. Human biomonitoring studies have shown decreased MEHP detection in recent years; however, sources of dolphin exposure to DEHP are likely different as dolphins have demonstrated significantly higher MEHP concentrations compared to human reference populations. Currently, the only restriction on phthalate use is a rule written by the Consumer Product Safety Commission (CPSC) enacted in April 2018. This rule permanently prohibits any children’s toy or childcare article from containing more than 0.1% of a series of phthalates, including DEHP, dibutyl phthalate (DBP), benzyl butyl phthalate (BBP), diisononyl phthalate (DINP), diisobutyl phthalate (DIBP), di-n-pentyl phthalate (DPENP), di-n-hexyl phthalate (DHEXP), and dicyclohexyl phthalate (DCHP) [106]. Some states have implemented restrictions providing further protections than the federal regulations. California, for example, has become the first state to ban DBP and DEHP from cosmetic and personal care product formulations [107]. This bill will not be completely enacted until 2025, so it is unclear how effective these measures may be in mitigating phthalate exposure. Consideration of further mitigation measures may be required to promote wildlife health.

## Figures and Tables

**Figure 1 animals-12-00824-f001:**
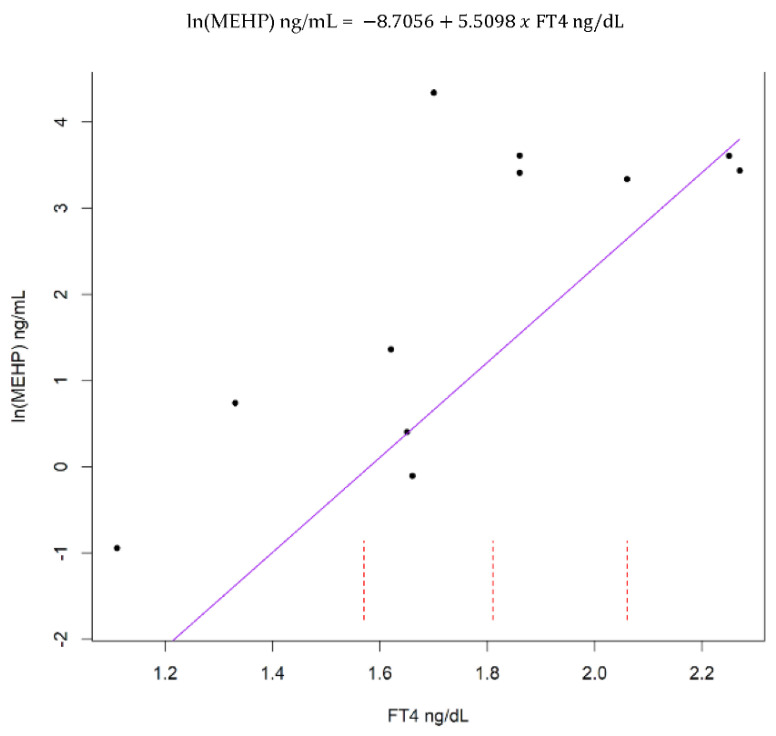
Akritas–Theil–Sen (ATS) line for the relationship between adult female free thyroxine (FT4) and natural log transformed mono(2-ethylhexyl) phthalate (MEHP). Kendall’s tau = 0.36, *p* = 0.04. Dashed red lines represent intervals for censored values below the limit of detection.

**Figure 2 animals-12-00824-f002:**
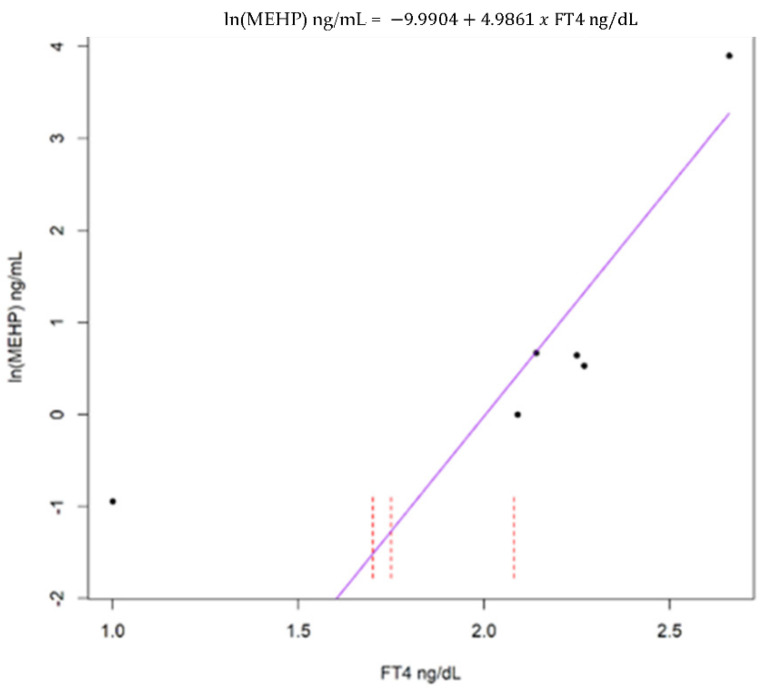
Akritas–Theil–Sen (ATS) line for the relationship between adult male free thyroxine (FT4) and natural log transformed mono(2-ethylhexyl) phthalate (MEHP). Kendall’s tau = 0.42, *p* = 0.02. Dashed red lines represent intervals for censored values below the limit of detection.

**Table 1 animals-12-00824-t001:** Descriptive statistics of mono(2-ethylhexyl) phthalate (MEHP) and thyroid hormone concentrations for individual dolphins (*n* = 50 unless otherwise specified) sampled from Sarasota Bay, Florida (2010–2019). Frequency of detection is given as %.

Analyte	Minimum	Median	Maximum	%
Mono(2-ethylhexyl) phthalate (MEHP; ng/mL)	0.17	0.49	76.60	56.00
Triiodothyronine (T3; ng/dL)	<LOD	86.55	175.00	60.00
Total thyroxine (T4; μg/dL)	7.94	15.00	22.02	100.00
Free thyroxine (FT4; ng/dL) (*n* = 49)	1.00	1.86	3.51	100.00

**Table 2 animals-12-00824-t002:** Comparison of mono(2-ethylhexyl) phthalate (MEHP) and thyroid hormone concentrations by sex for individual dolphins (*n* = 50 unless otherwise specified) sampled from Sarasota Bay, Florida (2010–2019). Frequency of detection given as %.

	Female (*n* = 29)				Male (*n* = 21)			
Analyte	Minimum	Median	Maximum	%	Minimum	Median	Maximum	%
Mono(2-ethylhexyl) phthalate (MEHP; ng/mL)	0.17	2.10	76.60	56.00	0.17	0.55	49.20	65.52
Triiodothyronine (T3; ng/dL)	<LOD	97.40	158.00	60.00	<LOD	70.30	175.00	65.52
Total thyroxine (T4; μg/dL)	10.55	15.80	22.02	100.00	7.94	14.20	21.10	100.00
Free thyroxine (FT4; ng/dL) (*n* = 49)	1.11	1.81	2.77	100.00	1.00	2.09	3.51	100.00

**Table 3 animals-12-00824-t003:** Comparison of MEHP and thyroid hormone concentrations by age class for individual dolphins (*n* = 50 unless otherwise specified) sampled from Sarasota Bay, Florida (2010–2019). Frequency of detection given as %.

	Adult (*n* = 33)				Juvenile (*n* = 17)			
Analyte	Minimum	Median	Maximum	%	Minimum	Median	Maximum	%
Mono(2-ethylhexyl) phthalate (MEHP; ng/mL)	0.17	0.42	76.60	52.94	0.17	1.28	28.40	62.50
Triiodothyronine (T3; ng/dL)	<LOD	80.55	158.00	58.82	<LOD	102.50	175.00	62.50
Total thyroxine (T4; μg/dL)	7.94	13.97	20.90	100.00	12.00	18.85	22.02	100.00
Free thyroxine (FT4; ng/dL) (*n* = 49)	1.00	1.70	2.66	100.00	1.22	2.14	3.51	100.00

**Table 4 animals-12-00824-t004:** Generalized linear model (GLM) results testing the association between hormone concentrations and demographic factors. Values in **bold** are significant at *p* < 0.05.

Hormone	Sex		Age Class		(Sex) *x* (Age Class)	
	Wald Statistic	*p*	Wald Statistic	*p*	Wald Statistic	*p*
Triiodothyronine (T3)	0.11	0.74	1.41	0.23	0.83	0.36
Total thyroxine (T4)	0.78	0.38	23.11	**<0.0001**	2.24	0.13
Free thyroxine (FT4)	5.96	**0.01**	14.56	**0.0001**	0.91	0.34

## Data Availability

Restrictions apply to the availability of these data. Dolphin hormone data were obtained from The Sarasota Dolphin Research Program (SDRP) and are available from the authors with the permission of SDRP. The phthalate data used for this study can be accessed through the 4TU.Research Data international data repository for science, engineering, and design Accessed 17 April 2021. https://doi.org/10.4121/14455782.

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
