# Peer review of "A Correlational Analysis of Phthalate Exposure and Thyroid Hormone Levels in Common Bottlenose Dolphins (Tursiops truncatus) from Sarasota Bay, Florida (2010–2019)"

_animals, 2022, doi:10.3390/ani12070824_

Round 1

Reviewer 1 Report

  1. This is a clear, well-written report that should be of interest to many physiologists and veterinarians.
  2. I found no serious errors or problems. I think the authors used appropriate methods, and they describe their methods and findings clearly and effectively. The manuscript anticipated and answered nearly all of my questions (for example, would there be seasonal fluctuations in circulating thyroid hormones). I have few suggestions for improving the manuscript, except for some things I list below.
  3. Maybe I am missing something, but I cannot see how Table 3 presents data by age class. It appears to show no results by age whatsoever.
  4. Captions for multiple tables indicate that values in bold are significant, but they do not show any values in bold type.
  5. The captions for the two figures include all relevant information, so the text above the plots (from the graphics program) can be deleted in both cases; it is unnecessary and distracting.
  6. The comma splice at the end of line 232 should be replaced with a semicolon or period—actually, that may be a semicolon there already. It is hard to see with the superscript numbers.
  7. The Discussion is quite interesting, especially the portions considering potential endocrine dysfunction in dolphins. I agree that “phthalate metabolism is likely different between humans and dolphins” (line 246). It might help the general reader to learn if there is relevant basic comparative information; for example, are thyroid glands notably smaller or larger relative to body size in dolphins as compared to humans? I was also curious to know if studies have found elevated levels of phthalates in fishes, so I was pleased to see zebrafish mentioned once here (line 265).
  8. I wonder if phthalate chemistry might naturally be somewhat different in seawater versus the freshwater that humans drink and are typically concerned about. Specifically, are there marine microbes (e.g., sulfur reducers) that degrade or otherwise alter/metabolize these chemicals (phthalates and their esters, etc.), such that the chemistry and/or environmental biology of phthalates is notably different in marine ecosystems than terrestrial ones?
  9. On a related note, I wonder how much seawater could increase exposure to phthalates because of easier breakdown/degradation of plastic containers and nurdles due to salinity, wave and other physical action, UV exposure, etc. Any reason to suspect that MEHP or DEHP is a bigger problem in the sea than for land mammals?

Reviewer 2 Report

Please provide additional information on serum analytes assays: method, reagents manufacturer and, most importantly, the lowest detectable concentration. It is crucial for reproducibility of the results making them more valuable for the future comparisons including long-term monitoring and regional differences.

If possible, specify pre-centrifugation time (range) since it affects hormone concentrations detected by immunoassays. Between them radioimmunoassay is less affected. (For instance, doi: 10.1258/acb.2007.007183)

Lines 144-145. The list of analytes includes aldosterone, progesterone, estradiol, and testosterone, but the article does not provide any relevant data.

Since the study deals with thyroid hormones, I think it should be indicated in the title.

Tables 3 and 4. No values in bold indicated.

When dealing with both total and free thyroxine they are usually referred to as total T4 and free T4 (but T4 and fT4). So, using «total T4» instead of «T4» seems more appropriate for this paper.

Reviewer 3 Report

Dear authors,

I do really think your publication is in a great value, and quite complete. However, I do believe there are some issues to be amended for publication.

Although authors have made an amazing review of the contaminants and their effects, I do think there are too many references. The manuscript is not a review manuscript (like a state of art); therefore references should be reduced.

If authors want to present all references as review part of the “State of art”, my suggestion is to compile that information in a table and include it in Supplementary material.

In addition, there are few bits that in my opinion should not be in the section that authors included. See my comments attached with the specific suggested changes.

Regards

Author Response

Please see attachment. Overall, approximately 20 citations were removed from the document. 
